# Quantitative Evaluation of the Infrazygomatic Crest Thickness in Polish Subjects: A Cone-Beam Computed Tomography Study

**Marta Gibas-Stanek** * , **Julia Ślusarska, Michał Urzędowski, Szczepan Żabicki and Małgorzata Pihut**

Department of Prosthodontics and Orthodontics, Dental Institute, Faculty of Medicine, Jagiellonian University Medical College, Montelupich St. 4/108, 31-155 Krakow, Poland; jula.slusarska@gmail.com (J.Ś.); michal.urzedowski@gmail.com (M.U.); szczepanzabicki@gmail.com (S.Ż.); malgorzata.pihut@uj.edu.pl (M.P.)
* Correspondence: marta.gibas-stanek@uj.edu.pl

**Abstract:** Infrazygomatic crest (IZC) mini-implants are frequently used as an absolute anchorage when intrusive or distally directed forces are required during orthodontic treatment. The aim of the present study was to evaluate the thickness of the IZC area in Polish patients as well as to assess dependency between bone availability, sex, and age. The study material was 100 cone beam computed tomography scans (CBCT) of the maxilla of patients of the University Dental Clinic in Krakow (50 men and women each). IZC bone thickness was measured at nine different points. The biggest bone thickness was recorded in the interdental space between the first and second molar at the height of 12 mm (6.03 ± 2.64 mm). The thinnest bone depth was localized at the level of the mesial root of the first molar, 16 mm above the occlusal plane (2.42 ± 2.16). There was a significant and negative correlation between bone thickness and age in the case of measurements taken buccally to the first molar. Only two out of nine measurements showed a sex dependency (points I2 and I3). Considering vertical and sagittal dimensions, the most favorable conditions for IZC mini-implant placement were found interdentally, between the first and second molar, 12 mm above the occlusal plane.

**Keywords:** bone screw; maxilla; orthodontics; X-ray computed tomography

## 1. Introduction

The infrazygomatic crest (IZC) is a thick bony ridge located between the first and the second maxillary molar. It has been used as a site for orthodontic miniplate placement when absolute anchorage or intrusive or distally directed forces were required during orthodontic treatment [1,2]. Hugo de Clerck, the author of the method, recommends this location due to its solid bone structure and safe distance from the roots of the upper molars [1]. The Zygoma Anchor System, designed and popularized by de Clerck, consists of a titanium miniplate adjusted to the shape of the infrazygomatic buttress and three miniscrews (5 or 7 mm long), that fixes the miniplate to the bone. Although miniplates provide effective orthodontic anchorage, the process of their placement requires advanced surgical skills. The procedure consists of preparation of the mucoperiosteal flap, bending of the miniplate to the proper shape, fixation with miniscrews, repositioning of the mucoperiosteum, and placing sutures. After completion of the orthodontic treatment, miniplates have to be removed. To simplify the clinical procedure and minimize the financial cost of orthodontic treatment, extra-alveolar IZC miniscrews have been introduced. They are placed on the buccal surface of the alveolar process at the base of the zygomatic crest eminence as an alternative to miniplates. Chris Chang, who popularized IZC, recommends its use for the retraction of posterior teeth and rotation of the whole dental arch in order to manage even challenging malocclusions without extractions or orthognathic surgery [3].

Despite their undeniable advantages, extra-alveolar miniscrews inserted in the IZC area can perforate the maxillary sinus and initiate sinus infection or mini-implant loss [4]. Although according to some authors [5], sinus penetration depth within 1 mm is advocated

for mini-implant anchorage, to avoid this complication, adequate bone depth should be available at the site of mini-screw insertion. Typically, IZC mini-screws are longer than intra-radicular mini-implants and manufactured in two dimensions: 2 mm diameter and 12 mm or 14 mm length (a 14 mm screw is recommended in the case of thick soft tissue in the buccal vestibule) [6]. As stated by the literature, a minimum of 6 mm of the bone is necessary to provide sufficient stability of the IZC during orthodontic treatment [7,8]. According to our clinical observations of Polish patients, in the vast majority of cases, the depth of bone ridge in this area is thinner, making IZC mini-screw placement challenging or even impossible. In the face of a lack of studies assessing the maxillary bone thickness for optimum IZC mini-screw placement in the Polish population, the aim of the present study was to evaluate the thickness of the IZC area.

It was found that the IZC is located between the maxillary second premolar and the first molar in youngs and at the level of the maxillary first molar in adults [9]. Liou developed a method for IZC screw placement adjacent to buccal surfaces of the maxillary first molar [7], while Lin recommends a more distal site that is buccal to the maxillary second molars [10,11]. Nevertheless, the region between the first and the second molar is also recommended for IZC screw placement [12]. To localize the most favorable site for IZC mini-screw implantation, we compared bone thickness at different levels considering sagittal plane and vertical distance from the occlusal plane. In view of the fact that both young and adult patients are seeking orthodontic treatment, another objective was to assess dependency between bone availability, sex, and age.

## 2. Materials and Methods

The study was approved by the bioethics committee of Jagiellonian University (number 1072.6120.132.2020).

### 2.1. Materials

The study material was 100 consecutively selected cone beam computed tomography scans (CBCT) of the maxilla of patients of the University Dental Clinic in Krakow (50 men and women each), taken for any reason between 2018 and 2021. Sample size determination showed that 91 patients would be suitable based on a margin of error of 0.05, confidence level 0.8, and population size of 200. Subjects included in the study met the following criteria: age of the patient >12 years, presence of upper first and second molar, absence of pathologies of the maxilla, and absence of clefts and maxillofacial syndromes. Patients with a history of orthodontic treatment, facial surgery, or facial trauma as well as images with artifacts were excluded from the research.

The CBCT scans were acquired with the OP 3D Pro (KaVo, Berlin, Germany). The protocol was: field of view $130 \times 150$ mm, average exposure time 8.5 s, average scanning time 39 s, average voxel size 380 $\mu m^{-5}$ mA. For the analysis of the CBCT images, a medical diagnostic monitor (RadiForce MX215, EIZO, Viena, Austria) and InViVo Dental Viewer (Anatomage, Santa Clara, CA, USA) were used. All the scans were analyzed by a single trained and calibrated senior postgraduate trainee in orthodontics.

### 2.2. IZC Bone Thickness Measurements

IZC bone thickness was measured at the level of the mesiobuccal and distobuccal root of the first permanent molar and between the first and second molar as presented in the Figure 1. To localize the buccal roots of the first molar, a horizontal view was used. In the sagittal view, the horizontal axis was located at the level of $\frac{1}{2}$ height of the roots. Subsequently, a picture of the maxilla in horizontal projection was rotated in relation to the coronal axis to obtain an angle of 90° between the coronal axis and buccal surface of the alveolar process at the level of point M (mesial buccal root of the first molar), point D (distal buccal root of the first molar), and point I (interdental space between the first and the second molar) to imitate clinical conditions during mini-implant placement. Using the occlusal plane as a reference line, three points in the vertical plane were defined in

modified coronal view at the levels of 12 mm, 14 mm, and 16 mm from the occlusal plane as points of potential mini-implants insertion. All the measurements were taken at an angle of 70 degrees to the occlusal plane, on the patients' left side.

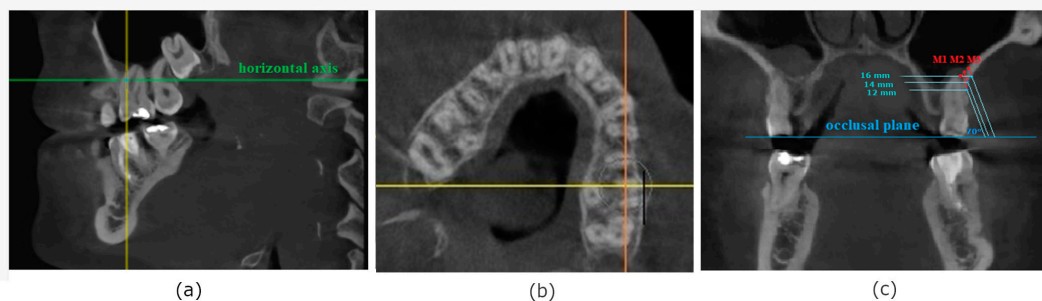

(a)                    (b)                    (c)

**Figure 1.** Measurement of bone depth on CBCT scans: (**a**)—sagittal slice consisting of mesiobuccal root of the first permanent molar used to obtain horizontal slice; (**b**)—horizontal view of the maxilla rotated in relation to the coronal axis to obtain a cross-section of the alveolar process at the level of M1; (**c**)—measurement of IZC thickness in modified coronal view.

### 2.3. Palatal Bone Thickness Measurements

Modified palatal height index was also calculated as a relation between the height and width of the palate according to the formula:

$$\text{Modified palatal height index } = \frac{\text{Palatal height } \times 100\%}{\text{Palatal width}}$$

For the purpose of this measurement, coronal slice of CBCT was selected, when the coronal axis was set at the center of the crown of the first molar. Although to calculate palatal height index [13] the distance between palatal cusps of the first maxillary molars should be used, for the purpose of our study, palatal width was defined as the horizontal distance between alveolar ridges on the palatal side of the right and left first maxillary molar to eliminate the influence of the inclination of these on the result (Figure 2). The perpendicular distance between the midpalatal raphe and the occlusal plane was measured to evaluate palatal height.

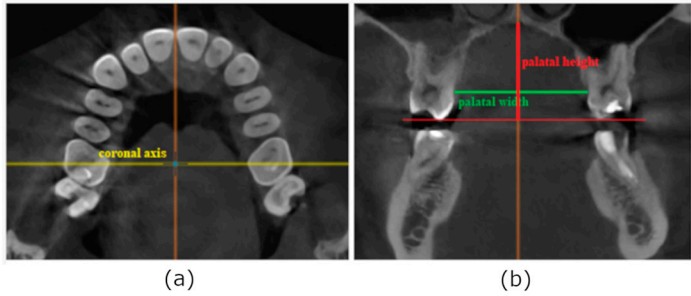

(a)                    (b)

**Figure 2.** Measurement of modified palatal height index: (**a**)—horizontal slice used to obtain coronal view, (**b**)—measurements of the palatal width and height.

### 2.4. Intra-Examiner Error Calculation

After 3 weeks, measurements of 10 randomly selected CBCTs were repeated to calculate intra-examiner error and determine method reliability.

### 2.5. Statistical Analysis

Data analysis was performed using R software version 4.3.0 (R: A language and environment for statistical computing. R Foundation for Statistical Computing, Vienna, Austria).

To test whether the sample fits normal distribution, the Shapiro–Wilk test was used. Analysis of quantitative variables was performed by calculating mean, standard deviation, median, and quartiles. Qualitative variables were analyzed by calculating the number and percentage of occurrences of each value. The Mann–Whitney test was used to compare quantitative variables between two groups, while the Kruskal–Wallis test (followed by Dunn post-hoc test) was used for more than two groups. Concordance of measurements of quantitative variables was assessed with ICC (Intraclass Correlation Coefficient) type 2 (according to the Shrout and Fleiss classification). The Friedman test (followed by paired Wilcoxon tests with Bonferroni correction as a post-hoc procedure) was used to compare more than two repeated measures of quantitative variables. The relationship between two quantitative variables was assessed with Spearman's coefficient of correlation.

The significance level for all statistical tests was set to 0.05.

## 3. Results

### 3.1. Patients Characteristics

A total of 100 consecutively selected CBCT images of the maxilla (50 men and women each) that met the inclusion criteria were evaluated in this study. The mean ages of women and men were 28.7 and 28.92, respectively. Patients' age details are presented in Table 1.

**Table 1.** Patients characteristics.

| Sex | N | Age | | | | | | | *p* |
| | | Mean | SD | Median | Min | Max | Q1 | Q3 | |
|---|---|---|---|---|---|---|---|---|---|
| Female | 50 | 28.70 | 13.83 | 25 | 12 | 65 | 16.5 | 38.75 | *p* = 0.654 |
| Male | 50 | 28.92 | 11.95 | 27 | 13 | 63 | 18.0 | 36.75 | |
| Total | 100 | 28.81 | 12.86 | 25 | 12 | 65 | 18.0 | 38.00 | |

*p*—Mann–Whitney test, SD—standard deviation, Q1—lower quartile, Q3—upper quartile.

### 3.2. Intra-Examiner Error Calculation

Within a 3-week interval, 10 randomly selected CBCT scans were subjected to repeated measurements. The concordance of the measurements of quantitative variable assessment with intraclass correlation coefficient type 2 (according to the Shrout and Fleiss classification) indicated good or excellent accordance between the first and second measurements (Table 2).

**Table 2.** Intra-examiner error calculation.

| Parameter | Measurement 1 (Mean ± SD) | Measurement 2 (Mean ± SD) | ICC | 95% CI | | Agreement (Cicchetti) | Agreement (Koo and Li) |
|---|---|---|---|---|---|---|---|
| M1 | 1.86 ± 2.29 | 1.88 ± 2.42 | 0.995 | 0.980 | 0.999 | Excellent | Excellent |
| M2 | 2.84 ± 2.77 | 2.89 ± 2.84 | 0.995 | 0.983 | 0.999 | Excellent | Excellent |
| M3 | 2.37 ± 2.13 | 2.79 ± 1.98 | 0.895 | 0.646 | 0.973 | Excellent | Good |
| D1 | 4.67 ± 3.15 | 4.48 ± 3.25 | 0.987 | 0.950 | 0.997 | Excellent | Excellent |
| D2 | 3.94 ± 2.29 | 3.87 ± 2.31 | 0.991 | 0.965 | 0.998 | Excellent | Excellent |
| D3 | 2.91 ± 1.65 | 2.86 ± 1.61 | 0.959 | 0.853 | 0.989 | Excellent | Excellent |
| I1 | 6.14 ± 1.34 | 6.01 ± 1.5 | 0.695 | 0.189 | 0.913 | Good | Fair |
| I2 | 5.24 ± 1.91 | 5.47 ± 2.14 | 0.911 | 0.700 | 0.977 | Excellent | Excellent |
| I3 | 4.52 ± 1.74 | 4.66 ± 1.7 | 0.974 | 0.904 | 0.993 | Excellent | Excellent |
| Height of the palate | 20.4 ± 2.53 | 20.42 ± 2.82 | 0.973 | 0.903 | 0.993 | Excellent | Excellent |
| Width of the palate | 33.91 ± 2.62 | 33.72 ± 2.75 | 0.965 | 0.873 | 0.991 | Excellent | Excellent |
| Palatal height index | 60.15 ± 5.83 | 60.53 ± 6.79 | 0.960 | 0.856 | 0.990 | Excellent | Excellent |

### 3.3. Bone Thickness

Tables 3–5 present differences in the mean bone thickness at the level of 12 mm, 14 mm, and 16 mm depending on the sagittal location of the mini-implant. At the height of 12 mm

and 14 mm, bone layer was the thickest in the interdental space between the first and the second molar (6.03 ± 2.64 mm and 4.74 ± 2.17 mm, respectively) followed by the area of the distal root of the first molar (3.71 ± 2.76 mm and 3.11 ± 2.35 mm). In the case of the measurements performed at the height of 16 mm, the difference was statistically insignificant.

**Table 3.** Mean bone thickness at the height of 12 mm from the occlusal plane.

| Bone Thickness (mm) | M1 | D1 | I1 | *p* |
|---|---|---|---|---|
| mean ± SD | 2.5 ± 2.55 | 3.71 ± 2.76 | 6.03 ± 2.64 | $p < 0.001$ |
| median | 2.13 | 3.53 | 5.94 | |
| quartiles | 0–3.9 | 1.53–5.23 | 4.25–7.65 | I1 > D1 > M1 |

*p*—Friedman test + post-hoc analysis (Wilcoxon paired tests with Bonferroni correction).

**Table 4.** Mean bone thickness at the height of 14 mm from the occlusal plane.

| Bone Thickness (mm) | M2 | D2 | I2 | *p* |
|---|---|---|---|---|
| mean ± SD | 2.54 ± 2.42 | 3.11 ± 2.35 | 4.74 ± 2.17 | $p < 0.001$ |
| median | 2.36 | 3.29 | 4.84 | |
| quartiles | 0–4.02 | 0.86–4.64 | 3.04–6.17 | I2 > D2 > M2 |

*p*—Friedman test + post-hoc analysis (Wilcoxon paired tests with Bonferroni correction).

**Table 5.** Mean bone thickness at the height of 16 mm from the occlusal plane.

| Bone Thickness (mm) | M3 | D3 | I3 | *p* |
|---|---|---|---|---|
| mean ± SD | 2.42 ± 2.16 | 2.59 ± 2.08 | 3.46 ± 1.93 | $p = 0.453$ |
| median | 2.16 | 2.29 | 3.54 | |
| quartiles | 0–3.95 | 1–3.65 | 1.83–4.7 | |

*p*—Friedman test.

Tables 6–8 and Figure 3 present differences in bone thickness depending on the vertical location of the mini-implant. In the case of the area of the mesial root of the first molar, there was no significant difference in the bone thickness at different levels. In the case of the area of distal root and interdental space, the greatest values of bone thickness were recorded at the level of 12 mm from the occlusal plane (3.71 ± 2.76 mm and 6.03 ± 2.64 mm, respectively) and with the growing distance from the occlusal plane, the bone thickness presented a decreasing trend.

**Table 6.** Mean bone thickness at the level of the mesial root of the first molar.

| Bone Thickness (mm) | M1 | M2 | M3 | *p* |
|---|---|---|---|---|
| mean ± SD | 2.5 ± 2.55 | 2.54 ± 2.42 | 2.42 ± 2.16 | $p = 0.098$ |
| median | 2.13 | 2.36 | 2.16 | |
| quartiles | 0–3.9 | 0–4.02 | 0–3.95 | |

*p*—Friedman test.

**Table 7.** Mean bone thickness at the level of the distal root of the first molar.

| Bone Thickness (mm) | D1 | D2 | D3 | *p* |
|---|---|---|---|---|
| mean ± SD | 3.71 ± 2.76 | 3.11 ± 2.35 | 2.59 ± 2.08 | $p < 0.001$ |
| median | 3.53 | 3.29 | 2.29 | |
| quartiles | 1.53–5.23 | 0.86–4.64 | 1–3.65 | D1 > D2 > D3 |

*p*—Friedman test + post-hoc analysis (Wilcoxon paired tests with Bonferroni correction).

**Table 8.** Mean bone thickness at the level of interdental space between the first and second molar.

| Bone Thickness (mm) | I1 | I2 | I3 | *p* |
|---|---|---|---|---|
| mean ± SD | 6.03 ± 2.64 | 4.74 ± 2.17 | 3.46 ± 1.93 | $p < 0.001$ |
| median | 5.94 | 4.84 | 3.54 | |
| quartiles | 4.25–7.65 | 3.04–6.17 | 1.83–4.7 | I1 > I2 > I3 |

*p*—Friedman test + post-hoc analysis (Wilcoxon paired tests with Bonferroni correction).

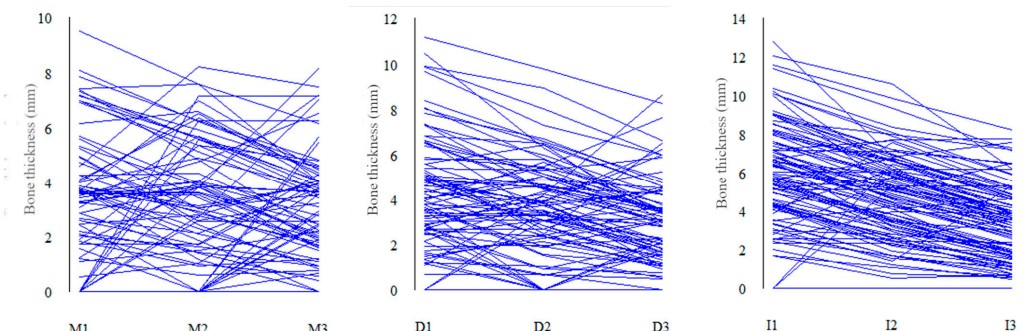

**Figure 3.** A parallel coordinate plot presenting bone thickness at the level of the mesial root of the first molar (M1, M2, M3), the distal root of the first molar (D1, D2, D3), and the interdental space between the first and the second molar (I1, I2, I3).

### 3.4. Correlation with Age

According to Spearman's correlation coefficient test, there was a significant and negative correlation between the bone thickness and the age in the case of measurement points M1, M2, M3, D1, D2, and D3 (Table 9). In points I1, I2, and I3 the correlation was also negative, but statistically insignificant. Results are also presented in Figure 4.

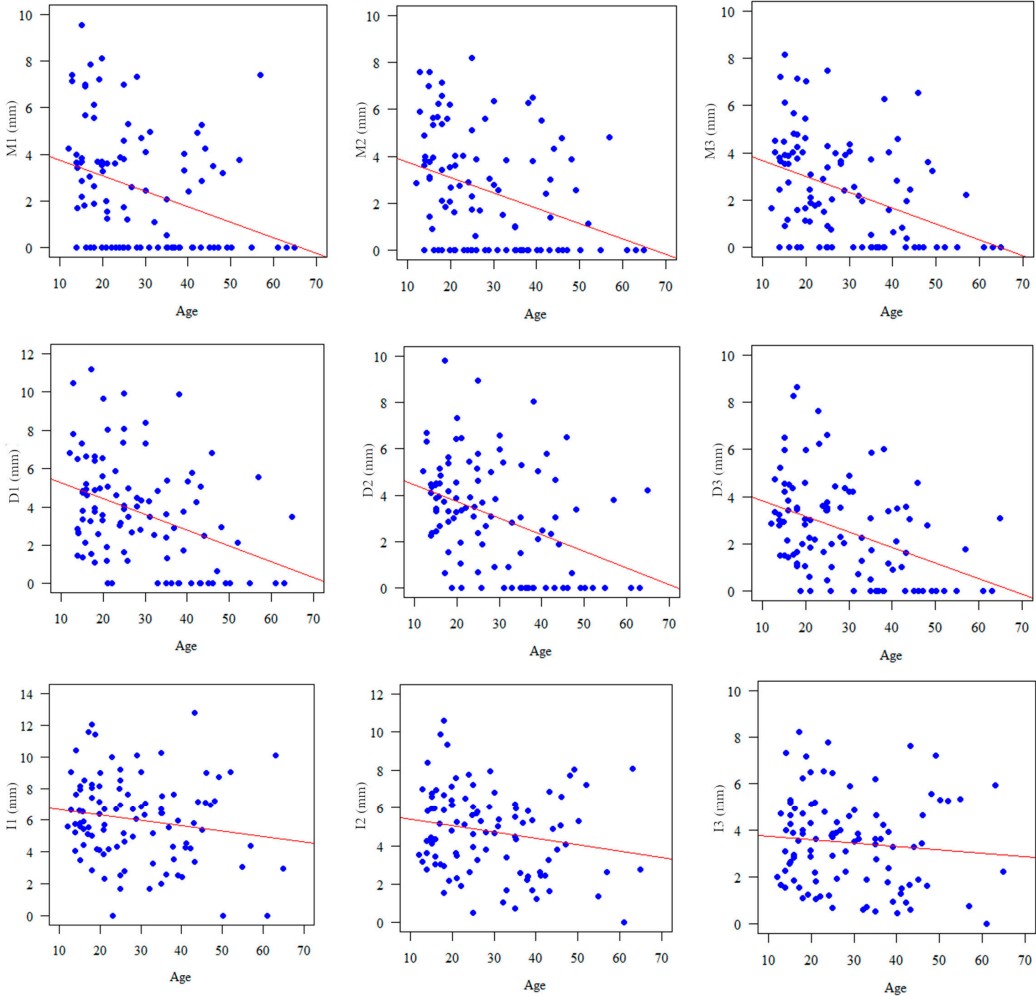

**Figure 4.** Correlation between the age of the patients and the bone thickness.

**Table 9.** Spearman's correlation coefficient test for comparison of parameters measured according to the age of the patients.

| Parameter | Age |
|---|---|
| | Spearman's Correlation Coefficient |
| M1 (mm) | $r = -0.37$, $p < 0.001$ * |
| M2 (mm) | $r = -0.381$, $p < 0.001$ * |
| M3 (mm) | $r = -0.424$, $p < 0.001$ * |
| D1 (mm) | $r = -0.378$, $p < 0.001$ * |
| D2 (mm) | $r = -0.406$, $p < 0.001$ * |
| D3 (mm) | $r = -0.42$, $p < 0.001$ * |
| I1 (mm) | $r = -0.147$, $p = 0.144$ |
| I2 (mm) | $r = -0.178$, $p = 0.077$ |
| I3 (mm) | $r = -0.107$, $p = 0.289$ |

* statistically significant ($p < 0.05$).

### 3.5. Correlation with Sex

Table 10 and Figure 5 report the correlation between the bone thickness and sex of the patients. The only statistically significant dependency was found at the I2 and I3 points, where bone thickness was greater in male patients.

**Table 10.** Correlation between the bone thickness and sex of the patients.

| Bone Thickness (mm) | Sex | N | Mean | SD | Median | Min | Max | Q1 | Q3 | p |
|---|---|---|---|---|---|---|---|---|---|---|
| M1 | Female | 50 | 2.96 | 2.77 | 3.29 | 0.00 | 9.53 | 0.00 | 4.86 | $p = 0.078$ |
| | Male | 50 | 2.03 | 2.25 | 1.73 | 0.00 | 7.86 | 0.00 | 3.60 | |
| M2 | Female | 50 | 2.65 | 2.16 | 2.71 | 0.00 | 7.59 | 0.15 | 4.00 | $p = 0.438$ |
| | Male | 50 | 2.42 | 2.68 | 1.48 | 0.00 | 8.21 | 0.00 | 4.03 | |
| M3 | Female | 50 | 2.32 | 1.96 | 1.96 | 0.00 | 7.20 | 0.79 | 3.78 | $p = 0.794$ |
| | Male | 50 | 2.52 | 2.36 | 2.42 | 0.00 | 8.15 | 0.00 | 4.06 | |
| D1 | Female | 50 | 3.54 | 2.64 | 3.50 | 0.00 | 9.87 | 1.40 | 5.05 | $p = 0.616$ |
| | Male | 50 | 3.88 | 2.90 | 3.74 | 0.00 | 11.18 | 1.97 | 5.36 | |
| D2 | Female | 50 | 3.00 | 2.12 | 3.04 | 0.00 | 8.04 | 1.62 | 4.20 | $p = 0.565$ |
| | Male | 50 | 3.22 | 2.57 | 3.42 | 0.00 | 9.79 | 0.66 | 5.03 | |
| D3 | Female | 50 | 2.44 | 1.83 | 2.14 | 0.00 | 6.51 | 1.06 | 3.50 | $p = 0.691$ |
| | Male | 50 | 2.74 | 2.31 | 2.82 | 0.00 | 8.64 | 0.78 | 4.06 | |
| I1 | Female | 50 | 5.53 | 2.40 | 5.61 | 0.00 | 10.10 | 4.18 | 7.52 | $p = 0.11$ |
| | Male | 50 | 6.52 | 2.80 | 6.50 | 0.00 | 12.80 | 4.95 | 8.02 | |
| I2 | Female | 50 | 4.14 | 2.09 | 3.94 | 0.00 | 8.01 | 2.51 | 5.91 | $p = 0.012$ * |
| | Male | 50 | 5.34 | 2.10 | 5.17 | 1.05 | 10.60 | 3.98 | 6.69 | |
| I3 | Female | 50 | 2.86 | 1.74 | 2.84 | 0.00 | 7.21 | 1.26 | 3.99 | $p = 0.003$ * |
| | Male | 50 | 4.06 | 1.94 | 3.86 | 0.48 | 8.22 | 2.58 | 5.28 | |

$p$—Mann–Whitney test, SD—standard deviation, Q1—lower quartile, Q3—upper quartile. * statistically significant ($p < 0.05$).

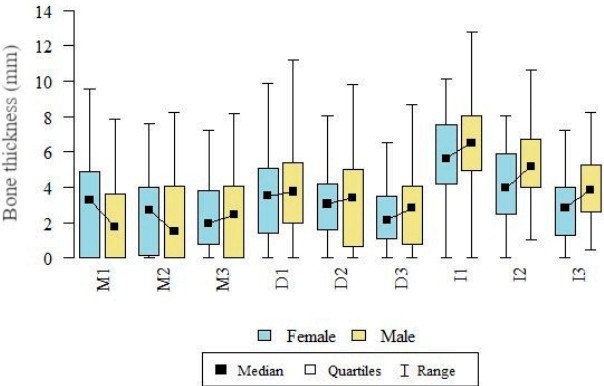

**Figure 5.** Graphic presentation of the correlation between the bone thickness and sex of the patients.

### 3.6. Correlation with Palatal Height Index

A significant and negative correlation between bone thickness and modified palatal height index was found in the case of points M1, D1, and D2 (Table 11, Figure 6). Nevertheless, it can be observed that bone depth tends to decrease with the increase of modified palatal index when areas buccal to the first molars are analyzed, and increase when interdental area is taken into consideration.

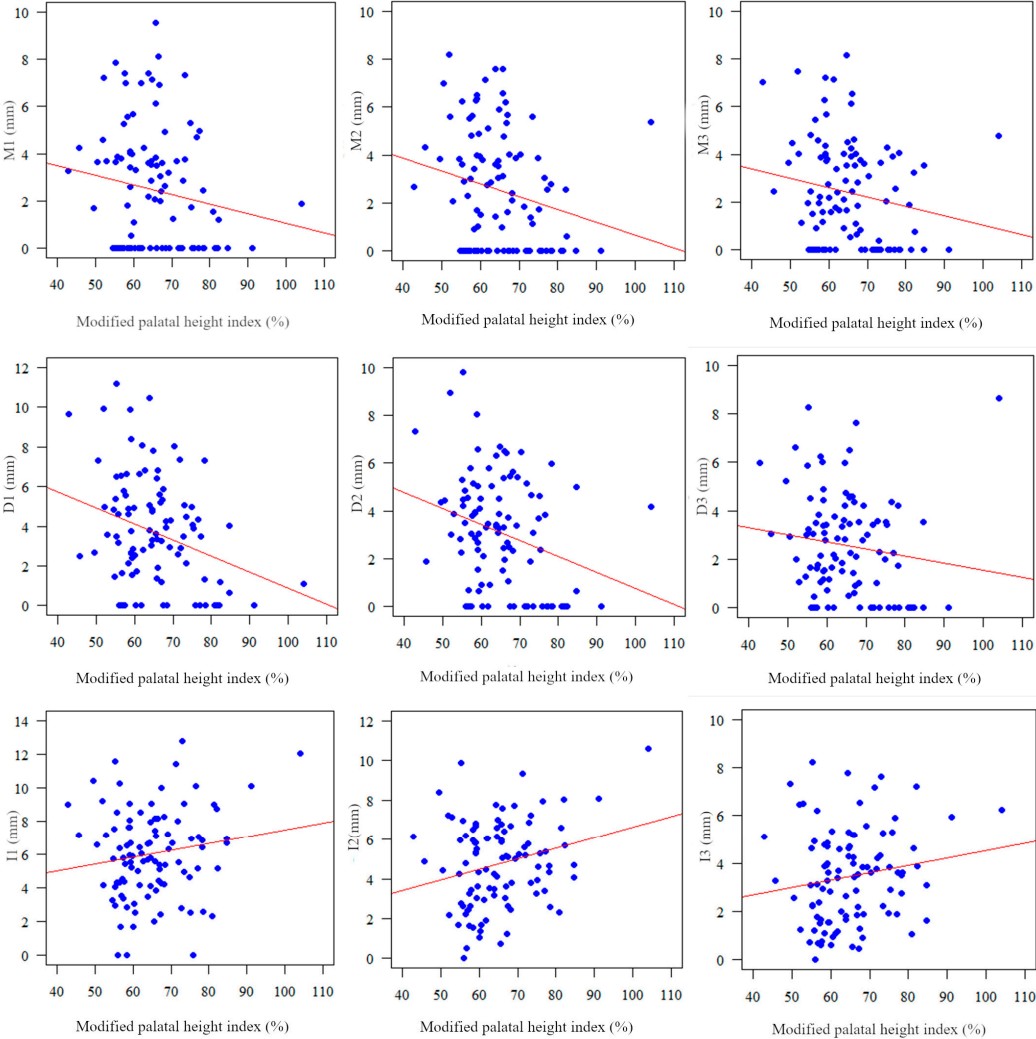

**Figure 6.** Correlation between the bone thickness and modified palatal height index.

**Table 11.** Correlation between the bone thickness and palatal height index.

| Parameter | Modified Palatal Height Index |
|---|---|
| | Spearman's Correlation Coefficient |
| M1 (mm) | $r = -0.113$, $p = 0.264$ |
| M2 (mm) | $r = -0.206$, $p = 0.04$ * |
| M3 (mm) | $r = -0.151$, $p = 0.134$ |
| D1 (mm) | $r = -0.208$, $p = 0.038$ * |
| D2 (mm) | $r = -0.249$, $p = 0.012$ * |
| D3 (mm) | $r = -0.184$, $p = 0.066$ |
| I1 (mm) | $r = 0.091$, $p = 0.369$ |
| I2 (mm) | $r = 0.175$, $p = 0.081$ |
| I3 (mm) | $r = 0.147$, $p = 0.146$ |

* statistically significant ($p < 0.05$).

## 4. Discussion

The area of the IZC is considered a safe zone for mini-screw placement, as this region is devoid of nerves, major blood vessels, or teeth [9]. Moreover, adequate bone density provides good primary stability, making IZC a valuable source of anchorage. Nevertheless, the close proximity of the maxillary sinus and the risk of Schneiderian membrane perforation, whose integration is vital for undisturbed sinus function [14], involves the need to accomplish the implantation procedure in a precise manner. As stated in the literature, when an implant invades the maxillary sinus less than 2 mm, the Schneiderian membrane is elevated and new bone is formed in this region. In the case of deeper insertion, the maxillary membrane becomes perforated, which may promote the development of sinusitis [15].

Theoretically, to obtain sufficient stability of the IZC mini-screw a minimum bone thickness of 6 mm is required [7]. Liou et al. report that the mean bone depth measured in Taiwan patients at the level of the mesial root of the first maxillary molar, 14 mm above the occlusal plane, at the insertion angle of 70° was 8 mm. According to our study, mean bone thickness at the same point and angle of insertion was 2.54 ± 2.42 mm, which implies much poorer conditions for implantation. Matias et al. conducted a study on Brazilian patients to identify the optimal areas for extra-alveolar mini-screws of patients with different facial patterns [16]. There was no significant difference in the case of IZC bone thickness among groups, but the mean bone depth at the level of the distal root of the first maxillary molar 13 mm above the occlusal plane and with the insertion angle of 70° ranged from 7.11 ± 1.95 mm in dolichofacial patients to 7.51 ± 2.16 mm in brachyfacial patients, which gives much higher results when compared to analogical measurements performed in our study (3.71 ± 2.76 mm at the height of 12 mm and 3.11 ± 2.35 mm at the height of 14 mm). Values obtained at the level of interproximal contact of the first and second maxillary molars in another Brazilian study (7.3 ± 3.0 mm) also surpass our results (6.03 ± 2.64 mm) [17]. Nevertheless, to confirm that the mean bone thickness of the IZC in Caucasian patients is lower when compared to the non-Caucasian population, multicenter studies in a much larger group of patients are needed.

The most favorable conditions for IZC mini-screws placement were found at the level of interdental space between the first and the second molar, where bone depth ranged from 3.46 ± 1.93 mm at the point I3 to 6.03 ± 2.64 mm at I1. The thinnest bone was identified at the area of the mesial root of the first molar (from 2.42 mm to 2.54 mm), which indicates that bone thickness increases from front to back. A study conducted by Amri et al. [18] on Arabian patients, according to which mean bone depth at the level of the mesial root of the first molar with an insertion angle of 70° was 3.90 mm, confirmed our conclusion that the available bone in this area is insufficient and the risk of sinus injury could be high.

According to Matias et al., mean bone depth tends to decrease with an increase in the distance from the occlusal plane (an increase of 2 mm in height was connected with a 2 mm decrease in the bone thickness) [16]. Although the results of our study seem to be in line

with the abovementioned theory, the decrease in bone depth with growing vertical distance from the occlusal plane is not so notable. The mean difference in bone depth between the insertion point at the height of 12 mm and 16 mm is 0.8 mm in the case of the area of the mesial root of the first molar, 1.12 mm at the area of the distal root of the first molar, and 2.57 mm in the interdental space.

Results of our study indicate the presence of a significant and negative correlation between the age of the patients and bone thickness buccally to the first molar. Similar correspondence can be also observed in the case of interdental point of insertion, but the results are not statistically significant. Only a few studies assessed age dependency. Amri et al. [18] did not notice age dependency in the case of insertion site between the first and the second molar among patients aged 18 to 42 years.

Considering differences in bone thickness among sexes, bigger values were observed in the case of male patients, but the only significant differences were found in the I2 and I3 measurements. Similarly, Santos et al. [19] and Amri et al. [18] did not find any differences in the bone thickness buccally to the distal and mesial root of the maxillary first molar respectively, comparing sexes.

Since there are studies where some correlation between facial pattern and bone availability at the IZC area has been found [20], the aim of our study was also to observe dependency between the shape of the maxilla, expressed as a modified palatal index, and the structure of the IZC region. Although it needs to be emphasized that significant results were obtained only in the case of M2, D1, and D2 points, the bone thickness buccally to the first molar tends to decrease with the increase of modified palatal height index. Interestingly, the bone depth In the interdental space tends to grow slightly with the increase of the modified palatal height index. To conclude, a narrow or high maxilla might be associated with a thin buccal bone plate in the area of the first molar and superior implantation conditions between the first and second molar. A study by Tavares et al. partially supports our results [17]. They found that bone thickness between the first and the second molar was greater in dolichofacial patients when compared to meso- and brachyfacial individuals. On the other hand, Matias et al., in a similar study, did not observe significant differences in the IZC thickness among groups [16]. Inconsistencies in the above-mentioned investigations may result from the different methods of classification of patients to brachy-meso- and dolichofacial groups (based on the angle between SN and Go-Gn line or VERT index). It should be also accentuated that the above-mentioned measurements might be strongly affected by the morphology of the mandible. Consequently, the palatal index used in our study may provide more uninfluenced information regarding spatial conditions for the implantation of mini-screws.

Considering the practical implications of the abovementioned research, it can be stated that in Polish patients, the mean bone depth at the area of infrazygomatic ridge rarely reaches a minimum recommended value of 6 mm. Surprisingly, our clinical experience indicates a relatively good incidence of IZC mini-screws survival, even in the absence of bone of sufficient thickness (Figure 7 presents a CBCT scan of IZC mini-screw application in the case of small bone depth).

Some authors attribute this high success rate to the penetration through two cortical layers, which guarantees more than 1 mm of the cortical bone and improves the primary stability of mini-implants [5,7]. Nonetheless, insufficient bone depth might be attributed to the increased risk of maxillary sinus membrane rupture. As stated in the literature, sinus invasion during IZC mini-screws placement is relatively common. Although according to the Chinese study, 78% of mini-implants penetrated into the maxillary sinus [5], irritation of the Schneiderian membrane measured as the increase in its thickness was significant only when mini-implants penetrated into the sinus deeper than 1 mm. Minor, uncomplicated injuries of the maxillary sinus by miniscrews may regenerate spontaneously and orthodontic treatment interruption and mini-implant removal are not recommended [8,21].

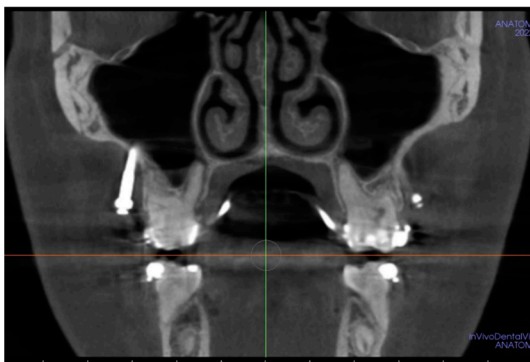

**Figure 7.** IZC mini-screw with an insertion depth of 1.5 mm, successfully used for the dystalization in the upper arch.

Although the findings of this study shed some light on IZC mini-screw placement, they have to be seen in light of certain limitations. One of them is the limited sample size coming from one center, so the results of the present research should not be extrapolated to the entire Polish population. Another limitation concerns the 380 μm voxel size—a smaller voxel size might provide even more precise measurements.

### 5. Conclusions

Considering sagittal and vertical dimensions, the most satisfactory conditions for IZC mini-screws placement were localized at the level of interdental space between the first and second molar, 12 mm above the occlusal plane, where mean bone depth reached $6.03 \pm 2.64$ mm. The thinnest bone depth was localized at the level of the mesial root of the first molar, 16 mm above the occlusal plane ($2.42 \pm 2.16$). There was a significant and negative correlation between bone thickness and age in the case of measurements taken buccally to the first molar. Due to the fact that individual variation in the growth and development of the maxilla and maxillary sinus may affect the anatomy of the IZC area, CBCT scan analysis should be considered prior to the mini-screw implantation procedure.

**Author Contributions:** Conceptualization, M.G.-S.; methodology, M.G.-S.; software, M.G.-S., J.Ś., M.U. and S.Ż.; validation, M.P.; formal analysis, M.G.-S., J.Ś., M.U., S.Ż. and M.P.; investigation, M.G.-S., J.Ś. and M.U.; resources, M.G.-S. and M.P.; data curation, M.G.-S.; writing—original draft preparation, M.G.-S.; writing—review and editing, J.Ś., M.U., S.Ż. and M.P.; visualization, M.G.-S.; supervision, M.P.; project administration, M.G.-S.; funding acquisition, M.G.-S. and M.P. All authors have read and agreed to the published version of the manuscript.

**Funding:** This research received no external funding.

**Institutional Review Board Statement:** The study was conducted according to the guidelines of the Declaration of Helsinki and approved by the bioethics committee of Jagiellonian University (number 1072.6120.132.2020).

**Informed Consent Statement:** Not applicable.

**Data Availability Statement:** The datasets generated and/or analyzed during the current study are available from the corresponding author upon reasonable request.

**Conflicts of Interest:** The authors declare no conflict of interest.

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
