# Peer review of "Quantitative Evaluation of the Infrazygomatic Crest Thickness in Polish Subjects: A Cone-Beam Computed Tomography Study"

_applsci, doi:10.3390/app13158744_

Round 1

Reviewer 1 Report

Dear Authors, first of all I would like to congratulate You on your work. The topic Is of great clinical relevance. However, I believe that the article could be improved. Please, take a note of some suggestions.

The abstract needs to be re-structured by shortening the methodology and adding highlighted results and a clear conclusion/ outcome of the study.

The methodology needs to be revised in a more organized way by adding a sub-heading for each step for clear understanding.

The picture/ illustration of the methodology should be expressed in a single composite image/ illustration following the sequence.

The conclusion section needs to be revised with a more clear and summarized outcome of the study

Author Response

Dear Reviewer,

Thank you for giving me the opportunity to submit a revised draft of my manuscript titled “Quantitative evaluation of the infrazygomatic crest thickness in Polish subjects. A cone-beam computed tomography study” to Applied Sciences. I appreciate the time and effort that you have dedicated to providing your valuable comments on my paper. I have been able to incorporate changes to reflect most of the suggestions. I have highlighted the changes within the manuscript and here is a point-by-point response to your comments and concerns.

Comments from Reviewer 1

Comment 1. The abstract needs to be re-structured by shortening the methodology and adding highlighted results and a clear conclusion/ outcome of the study.

Response: Thank you for this suggestion. I have rearranged the abstract. It can be found in the revised version of the manuscript. I also enclose the corrected version below:

Infrazygomatic crest (IZC) mini-implants are frequently used as an absolute anchorage when intrusive or distally-directed forces are required during orthodontic treatment. The aim of the present study was to evaluate the thickness of IZC area in Polish patients as well as to assess dependency between bone availability, sex, and age. The study material was 100 cone beam computed tomography scans (CBCT) of the maxilla of patients of the University Dental Clinic in Krakow (50 men and women each). IZC bone thickness was measured at nine different points. The biggest bone thickness was recorded in the interdental space between the first and second molar at the height of 12 mm (6.03±2.64 mm). The thinnest bone depth was localized at the level of the mesial root of the first molar, 16 mm above the occlusal plane (2.42±2.16). There was a significant and negative correlation between bone thickness and age in the case of measurements taken buccally to the first molar. Only two out of nine measurements showed a sex dependency (points I2 and I3). Considering vertical and sagittal dimensions, the most favorable conditions for IZC mini-implant placement were found interdentally, between the first and second molar, 12 mm above the occlusal plane.

 Comment 2. The methodology needs to be revised in a more organized way by adding a sub-heading for each step for clear understanding.

Response: Thank you for pointing this out. I added sub-headings in the “Material and Methods” section to make it more understandable.

Incorporated changes can be found in the main text on pages 2 and 3.

Comment 3. The picture/ illustration of the methodology should be expressed in a single composite image/ illustration following the sequence.

Response: Figures 2 and 3 are single composite images. To make it more understandable I added extra sentences in figures captions. Changes can be found within the revised manuscript on page 3.

Comment 4. The conclusion section needs to be revised with a more clear and summarized outcome of the study

Response: Thank you for your comment on that. The corrected version of this section can be found below and in the revised manuscript on page 11.

Considering sagittal and vertical dimensions, the most satisfactory conditions for IZC mini-screws placement were localized at the level of interdental space between the first and second molar, 12 mm above the occlusal plane, where mean bone depth reached 6.03±2.64 mm. The thinnest bone depth was localized at the level of the mesial root of the first molar, 16 mm above the occlusal plane (2.42±2.16). There was a significant and negative correlation between bone thickness and age in the case of measurements taken buccally to the first molar. Due to the fact that individual variation in the growth and development of the maxilla and maxillary sinus may affect the anatomy of the IZC area, CBCT scans analysis should be considered prior to mini-screw implantation procedure.

I look forward to hearing from you in due time regarding our submission and to respond to any further questions and comments you may have.

Sincerely

Marta Gibas-Stanek

Reviewer 2 Report

The article is written accurately and has innovation.

Author Response

Comment: The article is written accurately and has innovation.

Response:

Dear Reviewer,

Thank you for your positive opinion of my work and for the opportunity to submit a revised draft of my manuscript titled “Quantitative evaluation of the infrazygomatic crest thickness in Polish subjects. A cone-beam computed tomography study” to Applied Sciences. I appreciate the time and effort that you have dedicated to reviewing my paper.

Sincerely

Marta Gibas-Stanek

Reviewer 3 Report

Dear authors,

With interest I have read your article. The research aims to investigate quantity of bone infra-zygomatic crest available for mini screws at different insertion points and levels which is an interesting point for clinicians and researchers. The presentation is of high quality, below are few points to be dressed:

Introduction:  The sentence “ Postsurgical edema has been observed in the first several days as a common complication” seems to be excess as does not follow the narrative of the introduction and not supported by a citation.

Methods: For sample size calculations, why the target population was set to 200 if the study aims to target Polish population?

Methods: Please add a reference for measuring palatal width and height on CBCT

Discussion:  Line 310. Change “rapture“ into “rupture” or “perforation”

Author Response

Dear Reviewer,

Thank you for giving me the opportunity to submit a revised draft of my manuscript titled “Quantitative evaluation of the infrazygomatic crest thickness in Polish subjects. A cone-beam computed tomography study” to Applied Sciences. I appreciate the time and effort that you have dedicated to providing your valuable comments on my paper. I have been able to incorporate changes to reflect most of the suggestions. I have highlighted the changes within the manuscript and here is a point-by-point response to your comments and concerns.

Comment 1. Introduction:  The sentence “ Postsurgical edema has been observed in the first several days as a common complication” seems to be excess as does not follow the narrative of the introduction and not supported by a citation.

Response: Thank you for pointing it out. I agree, that this sentence does not match the context. I removed the sentence in the revised version of the manuscript.

Comment 2. Methods: For sample size calculations, why the target population was set to 200 if the study aims to target Polish population?

Response: Thank you for your comment on that. Indeed, we had some concerns regarding ‘population size’ for the sample size calculation, and finally, we decided to use the number of CBCT scans taken in the University Dental Clinic in Krakow between 2018 and 2021.

Comment 3. Methods: Please add a reference for measuring palatal width and height on CBCT

Response: Thank you for your comment on that. In our study, we modified palatal height index to eliminate the influence of the palato-buccal inclination of the first molars on the result (the measurement was performed between alveolar ridges, instead of palatal cusps of the first molars). I modified the description of the method and added a reference to the palatal height index. The incorporated changes can be found in the revised manuscript on page 3.

Comment 4. Discussion:  Line 310. Change “rapture“ into “rupture” or “perforation”

Response: Thank you for noticing this mistake.

I look forward to hearing from you in due time regarding our submission and to respond to any further questions and comments you may have.

Sincerely

Marta Gibas-Stanek